# Long-Term Visual Prognosis of Patients Following Lens-Sparing Vitrectomy for Stage 4A Retinopathy of Prematurity

**DOI:** 10.3390/ijms24032416

**Published:** 2023-01-26

**Authors:** Chiharu Iwahashi, Tomoki Kurihara, Kazuki Kuniyoshi, Shunji Kusaka

**Affiliations:** Department of Ophthalmology, Faculty of Medicine, Kindai University, Osakasayama 589-8511, Japan

**Keywords:** retinopathy of prematurity, vitrectomy, visual acuity, refractive error, myopia, lens-sparing vitrectomy, laser, anti-vascular endothelial growth factor

## Abstract

This study evaluated the long-term visual outcomes of patients in whom at least one eye underwent successful lens-sparing vitrectomy (LSV) for stage 4A retinopathy of prematurity (ROP). A retrospective chart review was conducted using the data of 61 eyes of 42 patients with a minimum 4-year follow-up after successful LSV, with or without anti-vascular endothelial growth factor (VEGF) therapy, and whose best-corrected visual acuity (BCVA) was measurable using Landolt rings at the final visit. The mean age at the final follow-up was 10.1 ± 3.3 years. Before LSV, all eyes underwent laser ablation therapy. Twenty eyes (32.8%) with high vascular activity received anti-VEGF therapy before LSV. The mean decimal BCVA at the final follow-up was 0.23 ± 0.26 (range: hand motion to 1.2). Twenty-three eyes (54.1%) had a decimal BCVA of ≥0.4. Among 49 phakic eyes at the final examination, the mean refractive error was −10.1 ± 5.0 D, with 37 eyes (75.5%) having high myopia (>−6.0 D). No significant differences were observed in terms of decimal BCVA and refractive errors between eyes with and without anti-VEGF therapy. Approximately half of the patients had a decimal BCVA of ≥0.4, despite myopic refraction after successful LSV for stage 4A ROP. LSV for stage 4A ROP seemed to be associated with good visual function, despite myopic refraction.

## 1. Introduction

Retinopathy of prematurity (ROP), which is caused by an abnormal development of retinal vessels in preterm patients [1], is the leading cause of infant blindness in both developed and developing countries [2]. Over the past few decades, the standard treatment for avascular immature retinas has been laser ablation in patients with treatment-requiring ROP [3]. Recently, the use of intravitreal injections of anti-vascular endothelial growth factor (anti-VEGF) agents such as bevacizumab (Avastin; Genentech Inc., South San Francisco, CA), ranibizumab (Lucentis; Genentech Inc.), and aflibercept (Eylea; Regeneron, Tarrytown, NY) has gained prominence [4,5,6,7]. With anti-VEGF therapy, the number of patients who develop tractional retinal detachment appeared to be decreasing. However, some eyes that show poor response to laser and/or anti-VEGF therapy may develop retinal detachment and require surgical intervention [8,9,10,11].

Lens-sparing vitrectomy (LSV) has been effective in achieving good surgical outcomes in patients with stage 4 ROP. Previous reports have demonstrated that the anatomical success rate of LSV for stage 4A ROP ranges from 84% to 100% [12,13,14,15,16]. Sparing the lens results in advantages in visual rehabilitation by avoiding aphakic amblyopia [12,17,18]. The mean postoperative Snellen visual acuity (VA) in stage 4A ranged from 20/550 to 20/58 [19,20,21,22]. Among those reports, the longest follow-up results at an average age of 7.1 years were reported by Singh et al. [21], who described a mean Snellen VA of 20/151, while three eyes had light perception or no light perception. Because children with a history of ROP treatment often have cognitive, motor, and language delays, VA measured at younger ages, i.e., 3–5 years, may not be reliable, and better VA can be measured at older ages. Therefore, it is important to evaluate postoperative VA with a long-term follow-up. Furthermore, most studies conducted to date on postoperative VAs after LSV reported their results before the introduction of anti-VEGF therapy. As anti-VEGF therapy is sometimes performed before LSV, it would be interesting to evaluate VA after LSV for ROP during the anti-VEGF therapy era.

The aims of this study were to investigate the long-term visual prognosis of patients following LSV for stage 4A ROP and to explore whether visual prognosis depends on the status of the fellow eye.

## 2. Results

### 2.1. Patient Demographics

A total of 42 patients (26 female children and 16 male children) who received LSV in at least one eye for stage 4A ROP were included in the analysis. The mean gestational age of the patients was 25.1 ± 1.9 weeks (range: 22.0–30.6 weeks) and the mean birth weight was 681.9 ± 234.7 g (range: 347–1658 g). The mean age at the final follow-up was 10.1 ± 3.3 years (range: 4.3–16.1 years). Before LSV, all eyes of the patients in this study had undergone laser ablation therapy at the referring hospitals. The mean postmenstrual age (PMA) at LSV was 41.4 ± 5.9 weeks (range: 34.9–67.7 weeks). Among the patients, 13 received LSV in one eye and did not undergo vitrectomy in the other eye (group 1); 21 received LSV in both eyes (group 2); and 8 received LSV in one eye, and vitrectomy and lensectomy in the other eye (group 3). The fellow eyes of all 13 patients of group 1 had been treated with laser ablation. Table 1 shows the demographic characteristics of each group. No significant differences were observed in the parameters among the three groups.

### 2.2. Refractive Errors and VA at the Final Follow-Up

Table 2 presents the refractive errors and VA of the patients at the final follow-up. The mean decimal best-corrected visual acuity (BCVA) at the final follow-up was 0.23 ± 0.26 (range: hand motion to 1.2). There were no significant differences among the three groups regarding PMA at LSV. In total, 4 eyes (30.1%) of group 1, 24 eyes (60.0%) of group 2, and 5 eyes (62.5%) of group 3 had a decimal BCVA of ≥0.4. Although not statistically significant, a higher proportion of eyes of groups 2 and 3 achieved a decimal BCVA of ≥0.4 than those of group 1.

During the follow-up period, 12 of 61 eyes (19.7%) underwent cataract surgery for the treatment of lens opacity. Among 49 phakic eyes at the final examination, 37 (75.5%) had high myopia of greater than −6.0 D. The mean refractive error of 49 phakic eyes was −10.1 ± 5.0 D at the final follow-up. No significant differences were observed in terms of refractive error among the three groups (*p* = 0.958).

### 2.3. Anti-VEGF Therapy

Among the 61 eyes, 20 eyes (32.8%) with high vascular activity received anti-VEGF therapy before LSV. Moreover, 15 eyes received intravitreal bevacizumab (IVB) injection, while 5 eyes received intravitreal ranibizumab (IVR) injection at 2–9 days before LSV. All eyes presented regression of tortuosity and dilation of the retinal vessels and tunica vasculosa lentis after anti-VEGF therapy. No ocular complications related to anti-VEGF therapy were detected. There were no significant differences in the use of anti-VEGF agents among the three groups. The decimal BCVA and refractive errors at the final follow-up showed no differences between eyes that received anti-VEGF therapy and those with no history of anti-VEGF therapy (Table 3).

### 2.4. Comparison of Refractive Errors and VA of Both Eyes of Each Patient

Figure 1 shows a comparison of the BCVA of eyes that underwent LSV and fellow eyes at the final follow-up among the three groups. Among the 13 patients of group 1, the mean decimal BCVA of eyes that underwent LSV was significantly worse than that of fellow eyes that did not require vitrectomy (0.10 ± 0.15 vs. 0.88 ± 0.75; *p* = 0.0005). Among the eight patients of group 3, the mean decimal BCVA of eyes that underwent LSV was significantly better than that of fellow eyes that underwent vitrectomy and lensectomy (0.30 ± 0.40 vs. 0.02 ± 0.08; *p* = 0.005).

The mean decimal BCVA of eyes of groups 1, 2, and 3 were 0.60 ± 0.43, 0.38 ± 0.38, and 0.29 ± 0.56, respectively. Figure 1b shows a comparison of the BCVA of eyes that received anti-VEGF therapy before LSV at the final follow-up among the three groups. The mean decimal BCVA of eyes of group 2 was significantly better than that of group 1 (*p* = 0.042); however, the number of cases in each group was small.

In group 1, the mean refractive error of eyes that underwent LSV was significantly greater than that of fellow eyes that did not require vitrectomy (−10.5 ± 5.3 D vs. −4.1 ± 4.0 D; *p* = 0.009).

Regarding whether the different treatment between the two eyes determines the better-seeing eye, three patients had approximately equal BCVA values in both eyes in group 1. Of the remaining 10 patients, the eyes after LSV had worse BCVA values than the fellow eyes that did not require vitrectomy. In group 2, 5 patients had approximately equal BCVA values in both eyes, and the remaining 16 patients showed a difference in BCVA values between the two eyes. In group 3, three patients had approximately equal BCVA values in both eyes, four patients had better BCVA values in the eyes after LSV than the fellow eyes after vitrectomy and lensectomy, and the remaining one patient had better BCVA in the eye after vitrectomy and lensectomy. To summarize, the eyes that underwent LSV tended to have worse vision than the fellow eyes that did not undergo vitrectomy, and had better vision than the fellow eyes that underwent vitrectomy and lensectomy.

## 3. Discussion

This study investigated the long-term visual outcomes of patients following successful LSV for stage 4A ROP in the era of anti-VEGF therapy in a real-world clinical setting in Japan. The mean decimal BCVA and refractive errors of the patients at a mean age of 10.1 years were 0.23 and −10.1 D, respectively. Approximately half of the patients had a decimal BCVA of ≥0.4, despite myopic refraction.

Infants with ROP tend to have more frequent cognitive, motor, and language delays than those without ROP [23]. Even in normal preschool children, the VA test performance improves with age [24]. Therefore, in this study, to evaluate postoperative BCVA more accurately, we had a long observation period with a mean age of 10.1 ± 3.3 years at the final follow-up. It was possible to measure BCVA at later time points in an increasing number of children (data not shown). When reviewing long-term visual outcomes following LSV for stage 4A ROP, two studies had reported a mean logMAR BCVA of 0.92–1.44, measured using Early Treatment Diabetic Retinopathy Study Charts or Pesando tests in children aged approximately 7 years [21,22]. In the present study, the converted mean logMAR BCVA at the final follow-up was 0.64, which appears to be relatively better than that of previous reports. This might be because this study focused on only those patients who achieved retinal attachment, and also because the mean age at the time of VA measurement was greater than that of previous reports. Moreover, the lack of standardization in VA measurement in the previous studies [21,22] might influence the results. We measured the BCVA of all patients using Landolt rings, which is the gold standard method of VA measurement in Japan [21,22].

There are concerns regarding the use of anti-VEGF therapy for ROP. Several studies have reported that anti-VEGF agents escape from the vitreous into the systemic circulation and suppress serum VEGF for weeks to months following IVB or IVR [25,26,27]. This prolonged suppression of systemic VEGF in the early newborn period may potentially affect neurodevelopmental growth. Studies on neurodevelopmental outcomes have reported conflicting results. Some retrospective studies have demonstrated a negative impact of anti-VEGF therapy on neurodevelopment [28,29], whereas other prospective small case studies have reported no differences in neurodevelopmental outcomes between infants treated with anti-VEGF agents versus those who received laser therapy [30,31]. In this study, we also found no difference in long-term visual outcomes and refractive errors between 20 eyes that received IVR or IVB and 41 eyes that did not receive anti-VEGF therapy. Although there may be no obvious adverse effect of anti-VEGF therapy on visual development, other possible systemic effects remain to be investigated. Currently, prospective randomized trials, namely, the RAINBOW extension study and the FIREFLEYE next study, are underway, which will reveal the long-term impact of anti-VEGF therapy on neurodevelopment in the future.

In this study, the mean refractive error of 49 phakic eyes at the final follow-up was −10.1 ± 5.0 D, with 75.5% eyes having a high myopia of greater than −6.0 D. Although there are several reports on VA after LSV for stage 4A ROP, there are only three previous reports on refractive errors in children aged 2–4 years. Holtz et al. reported an average refractive error of −6.78 D for nine eyes after LSV in children with an average of 3.9 years [32]. Agarkar et al. reported on the changes in myopia over a period—with the average refractive errors being −4.36, −5.09, −7.14, −9.13, and −7.47 D at the ages of 2 months, 6 months, 1 year, 1.5 years, and 2 years, respectively—and found that myopia increased with age [33]. Macor et al. reported an average refractive error of −11.25 D for 10 eyes after LSV in children with an average of 3.0 years [34]. The degree of myopia in our study was similar to, or greater than, that of previous reports. The age at examination was greater than that of previous reports, which may explain this difference in refractive outcomes.

In the present study, among patients of group 1 who underwent LSV in one eye and did not undergo vitrectomy in the other eye, the mean refractive error of the study eyes that underwent LSV was significantly greater than that of the fellow eyes treated with a laser. Macor et al. also reported similar results, wherein among four patients who underwent LSV for progressive stage 4A, with aggressive ROP in one eye and laser therapy in the other eye, the eyes that underwent LSV developed high myopia compared with the fellow eyes at a mean age of 3.1 years [34]. In contrast, Holtz et al. reported that infant eyes that underwent LSV for stage 4A ROP developed less myopia than the fellow eyes treated with laser alone [32]. They described that the difference in myopia was due to the posterior displacement of the lens–iris diaphragm, with a smaller contribution from reduced corneal power. Although we did not examine ocular biometric data, differences in surgical technique, such as the extent of anterior vitreous resection and the location of the wound, might cause differences in the displacement of the lens–iris diaphragm. Moreover, prematurity is a causative factor of myopia associated with ROP [35]. The difference in the severity of ROP between eyes that underwent LSV and the fellow laser-treated eyes might influence the difference in the refractive outcomes. Further studies with larger cohorts are required to evaluate the role of LSV on the refractive outcomes.

Preoperative severity of ROP is believed to be most severe in eyes that undergo vitrectomy and lensectomy (the fellow eye of group 3), followed by LSV and no vitrectomy, which is consistent with the visual prognosis in each group. However, a comparison of the LSV-treated eyes in each group revealed that more eyes of groups 2 and 3 achieved a decimal BCVA of ≥0.4 than those of group 1, although the difference was not statistically significant. In particular, in eyes that received anti-VEGF therapy, the BCVA of patients in group 1 was significantly worse than that of patients in group 2. Visual development is affected by the visual function of the fellow eye. Hence, it is possible that the better-seeing eyes without vitrectomy in group 1 hindered the visual development in the operated eye. This indicates that the condition of the fellow eye may be a potential bias when examining the relationship between visual prognosis and various ocular biometric data, including macular anatomical changes, such as macular degeneration or dragging. It would be preferable to compare eyes that underwent the same treatment in both eyes in such a study.

There were several limitations in this study. First, some cases were excluded due to difficulty in visiting our hospital regularly, because most patients were outborn and referred to our hospitals for the surgical treatment of ROP from remote areas. The intellectual disabilities of several of our patients may have affected the measured VA. It is possible that although the mean age of our patients was 10.1 years, which is much greater than that of previous studies, the visual outcomes of this study are underestimated. Second, the dosing of IVB at 0.25 mg used in our study is not generalizable, as it was different from the most commonly used dosage of 0.625 mg. The strength of our study is the relatively large number of patients with a long follow-up period, including the eyes with a history of anti-VEGF therapy in the real-world clinical setting.

In conclusion, LSV for stage 4A ROP appears to be associated with good visual function, despite myopic refraction. Anti-VEGF therapy did not appear to affect the refractive outcome or visual function. Regular visual rehabilitation with adequate refractive correction is advisable for follow-up after vitrectomy for patients with stage 4A ROP.

## 4. Materials and Methods

### 4.1. Patients

The medical records of consecutive patients with stage 4A ROP who underwent successful LSV at Osaka University Hospital (Osaka, Japan) or Kindai University Hospital (Osakasayama, Japan) between February 2006 and February 2018 and were followed up at Kindai University Hospital were retrospectively reviewed. Patients who underwent lensectomy during reoperation for progressive ROP, did not achieve retinal attachment, or VA could not be evaluated at the final visit, were excluded.

Data collected from the charts included sex, gestational age at birth, birth weight, stage of ROP at LSV, PMA at LSV, anti-VEGF therapy, follow-up period, and refractive error and BCVA at the final visit. In addition to the refractive error and BCVA of the eyes that underwent LSV, those of the fellow eye were collected for comparison.

The study protocol and an off-label use of anti-VEGF agents were approved by the Institutional Review Board of Kindai University Hospital (#26-251, 26-252). The study adhered to the tenets of the Declaration of Helsinki.

### 4.2. Indication of Anti-VEGF Therapy

Fundus photographs and fluorescein angiograms were taken at the initial examination using a RetCam 3 digital fundus camera (Natus, San Carlo, CA, USA). ROP stage and zone were evaluated based on the International Classification of Retinopathy of Prematurity [36]. Before LSV, an anti-VEGF agent was injected for type 1 ROP with high vascular activity. The selection of IVB or IVR depended on the treatment period. Patients treated before June 2015 received the IVB injection (0.25 mg), whereas those treated after July 2015 received the IVR injection (0.25 mg). All parents or legal guardians were informed regarding efficacy and possible complications before the IVB or IVR injection, and they provided written informed consent. Anti-VEGF drugs were intravitreally injected under the supervision of a neonatal doctor using a 30-gauge needle 0.5–1.0 mm away from the limbus in the neonatal intensive care unit under topical anesthesia or, if necessary, under sedation.

### 4.3. Indication and Surgical Procedure of LSV

LSV was performed in eyes with progressive stage 4A ROP. The surgical techniques of LSV first applied by Maguire and Trese in infants [17] were modified as previously described [37]. Briefly, after conjunctival peritomy, sclerotomies were performed 0.5–1 mm away from the limbus. To avoid lens damage, the direction of the sclerotomy was more posterior rather than toward the center of the eyeball [15]. A three-port vitrectomy using a 23-, 25-, or 27-gauge system was used. For fundus view, the wide-angle viewing system Resight^®^ (Carl Zeiss Meditec AG, Jena, Germany) was used during surgery. Fibrous tissue traction was released to achieve retinal reattachment. Membrane dissection using 23-, 25-, or 27-gauge horizontal and/or vertical scissors (DORC, Zuidland, The Netherlands) was minimized to avoid intraoperative bleeding and/or the formation of an iatrogenic retinal hole. In cases in which an iatrogenic retinal break was encountered, the membrane around the break was meticulously removed, followed by fluid/air exchange, laser photocoagulation on the retina around the break, and SF_6_ gas injection.

Lensectomy was performed for lens opacity that developed after successful LSV when the opacity had interfered with the visual axis. Cataract extraction with or without the implantation of an intraocular lens was performed by the aspiration technique. After the aspiration of the lens, a posterior circular and central manual capsulotomy was performed using capsulorhexis forceps. The intraocular lens was placed with the haptics in the bag and the optic captured behind the capsulorhexis to maintain the visual axis clearly, as described previously [38].

The patients were followed up for 1 month post-operation at our hospital, monthly to every 3 months at the referring hospitals, and then every year at our hospital.

### 4.4. VA Measurement

Cycloplegic refraction was tested in the patients after the instillation of cyclopentolate 1.0%, atropine 1.0%, or atropine 0.5% eye drops, while automatic refraction was tested using the auto refractometer (NIDEK ARK-530A; Nidek, Gamagori, Japan or Topcon KR-8100; Topcon Corporation, Tokyo, Japan) or the Righton Retinomax 3 Refract Keratometer (Right Group, Tokyo, Japan). VA was evaluated with suitable refractive correction using Landolt rings. Better- or worse-seeing eyes were defined on the basis of a previous report [39].

Contact lenses were used for optical correction in patients after performing lensectomy. The process and timing of visual rehabilitation were determined and implemented by consulting pediatric ophthalmologists at the referring hospitals.

### 4.5. Study Group

The study patients were divided into three groups. Group 1 included patients who underwent LSV in one eye and did not undergo vitrectomy in the fellow eye (13 eyes), group 2 included patients who underwent LSV in both eyes (40 eyes), and group 3 included patients who underwent LSV in one eye and vitrectomy and lensectomy in the other eye (8 eyes).

### 4.6. Statistical Analysis

Statistical analyses were conducted using the JMP version 14.0 for Windows (SAS Institute, Cary, NC, USA) software. Data are expressed as mean and standard deviation, unless otherwise stated. VA was converted into the logarithm of the minimum angle of resolution (logMAR) for data analysis. Based on a previous report [40], hand motions and light perception were assigned as 2.9 and 3.1 logMAR, respectively.

Non-normally distributed data were analyzed using the Mann–Whitney test (two groups). Continuous data were evaluated using the Kruskal–Wallis test (three groups), and categorical data were evaluated using the chi-square test. A paired *t*-test was used to compare refractive errors between the study eye and fellow eye. A p value of <0.05 was considered to indicate statistical significance.

## Figures and Tables

**Figure 1 ijms-24-02416-f001:**
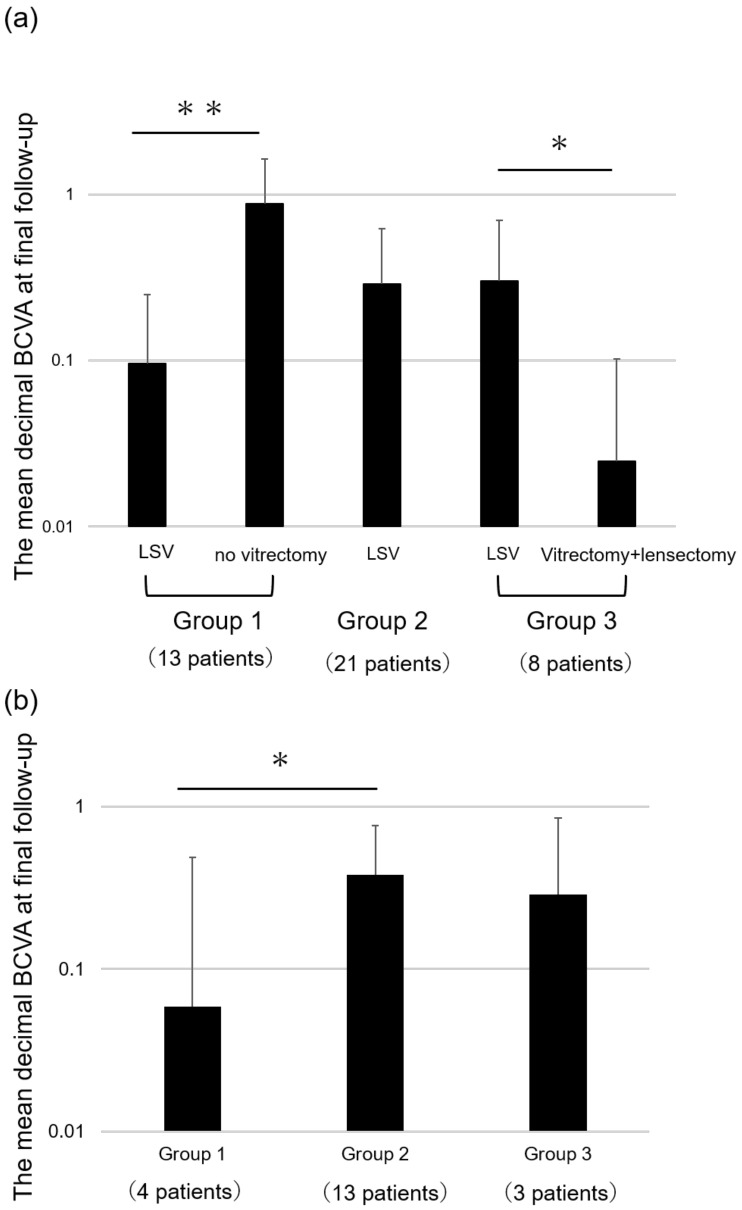
(**a**) Comparison between the mean decimal best-corrected visual acuity (BCVA) of eyes that underwent lens-sparing vitrectomy (LSV) and that of the fellow eyes at the final follow-up among the three groups. (**b**) Comparison of the mean BCVA of eyes that received anti-VEGF therapy before LSV at last follow-up among three groups. * *p* < 0.05, ** *p* < 0.001. Group 1—cases that underwent LSV but not vitrectomy in the other eye; group 2—cases that underwent LSV in both eyes; group 3—cases that underwent LSV in one eye and vitrectomy and lensectomy in the other eye. Data are presented as mean values with standard deviation.

**Table 1 ijms-24-02416-t001:** Demographic characteristics of patients who underwent successful lens-sparing vitrectomy for stage 4A retinopathy of prematurity.

	Group 1 (13 Patients)	Group 2 (21 Patients)	Group 3 (8 Patients)	Total (42 Patients)	*p* Value
Male children/female children	3/10	9/12	4/4	16/26	0.382
Gestational age (weeks)	24.8 ± 2.6	25.1 ± 1.6	25.3 ± 1.3	25.1 ± 1.9	0.433
Birth weight (g)	704.8 ± 358.4	644.1 ± 152.4	744.0 ± 162.9	681.9 ± 234.7	0.386
Age at the final follow-up (years)	10.8 ± 3.0	9.1 ± 3.2	11.7 ± 3.6	10.1 ± 3.3	0.134

Group 1—cases that underwent LSV but not vitrectomy in the other eye; group 2—cases that underwent LSV in both eyes; group 3—cases that underwent LSV in one eye and vitrectomy and lensectomy in the other eye.

**Table 2 ijms-24-02416-t002:** Ocular characteristics of eyes after successful lens-sparing vitrectomy for stage 4A retinopathy of prematurity.

	Group 1 (13 Eyes)	Group 2 (40 Eyes)	Group 3 (8 Eyes)	Total (61 Eyes)	*p* Value
PMA at LSV (weeks)	44.3 ± 10.5	40.7 ± 3.2	40.2 ± 5.8	41.4 ± 5.9	0.374
IVB/IVR	4/0	9/4	2/1	15/5	NA
Decimal BCVA at the final follow-up	0.10 ± 0.15	0.29 ± 0.33	0.30 ± 0.40	0.23 ± 0.26	0.202
0.4 or better	4 (30.1%)	24 (60.0%)	5 (62.5%)	33 (54.1%)	0.159
Refractive error at the final follow-up (D) #	−10.5 ± 5.3 (n = 11)	−9.8 ± 5.2 (n = 32)	−10.8 ± 4.4 (n = 6)	−10.1 ± 5.0 (n = 49)	0.958
Refractive error > −6.0D	9 (81.8%)	23 (71.9%)	5 (83.3%)	37 (75.5%)	0.794

Group 1—cases that underwent LSV but not vitrectomy in the other eye; group 2—cases that underwent LSV in both eyes; group 3—cases that underwent LSV in one eye and vitrectomy and lensectomy in the other eye. BCVA—best-corrected visual acuity; D—diopter; IVB—intravitreal injection of bevacizumab; IVR—intravitreal injection of ranibizumab; LSV—lens-sparing vitrectomy; PMA—postmenstrual age; #—cases that required lensectomy during the follow-up were excluded.

**Table 3 ijms-24-02416-t003:** Comparison of ocular characteristics between eyes that received anti-VFGF therapy and those with no history of anti-VFGF therapy.

	IVB/IVR (20 Eyes)	No Anti-VEGF Therapy (41 Eyes)	Total (61 Eyes)	*p* Value
Group 1/2/3	4/13/3	9/27/5	13/40/8	0.949
PMA at LSV (weeks)	39.7 ± 2.9	42.2 ± 6.9	41.4 ± 5.9	0.307
Decimal BCVA at the final follow-up	0.25 ± 0.32	0.22 ± 0.23	0.23 ± 0.26	0.963
0.4 or better	13 (65.0%)	20 (18.8%)	33 (54.1%)	0.230
Refractive error at the final follow-up (D) #	−10.8 ± 6.4 (n = 15)	−9.8 ± 4.4 (n = 34)	−10.1 ± 5.0 (n = 49)	0.467
Refractive error > −6.0D	11 (73.3%)	26 (76.5%)	37 (75.5%)	0.815

BCVA—best-corrected visual acuity; D—diopter; IVB—intravitreal injection of bevacizumab; IVR—intravitreal injection of ranibizumab; LSV—lens-sparing vitrectomy; PMA—postmenstrual age; #—cases that required lensectomy during the follow-up were excluded.

## Data Availability

The data presented in this study are available from the corresponding author on reasonable request.

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
