# Peer review of "Long-Term Visual Prognosis of Patients Following Lens-Sparing Vitrectomy for Stage 4A Retinopathy of Prematurity"

_ijms, 2023, doi:10.3390/ijms24032416_

Round 1

Reviewer 1 Report

Retinopathy of prematurity (ROP) is a retinal disease affecting babies that are born prematurely or are born with very low body weight. ROP is characterized by abnormal growth of blood vessels in the retina causing it to detach. Depending upon the severity of the disease, some may need surgery to prevent loss of vision. Several treatment options such as laser photocoagulation therapy, cryotherapy, lens- sparing vitrectomy (LSV) and anti-VEGF treatment have produced success with varying degrees. Long term prospective studies on these patients that have undergone surgery has been tough because testing visual acuity on younger patients is unreliable.

In this paper, Iwahashi et al., have performed a long-term follow-up study on young patients that were diagnosed with stage 4 ROP. Moreover, this paper is for the first time reporting a 4-year follow up analysis ROP patients that either received an anti-VEGF treatment prior to the LSV surgery or not. This paper is a good clinical paper that is written well and is a good addition to the existing knowledge about the postoperative patient visual health undergoing treatment for ROP. The authors acknowledge their limitations of the study but also provide a valuable suggestion for future work and future patient care for ROP patients. I have the following question:

1] Of the patients that received anti-VEGF treatment, which of the groups (1, 2, or 3) did they belong? And in Fig1, including data of mean decimal BCVA values of the subset of these patients that received anti-VEGF treatment before LSV will help understand the anti-VEGF treatment effect.

Reviewer 2 Report

The paper titled " Long-term visual prognosis of patients following lens-sparing vitrectomy for stage 4A retinopathy of prematurity" brings to the center of attention the management of a serious problem related to prematurity which has a significant impact on the quality of life. The introduction provides the necessary background for the understanding of the subject and most relevant references in the field are cited. The methodology is described in detail, the results are presented suggestively and the discussion is conducted in a comprehensive and logic manner. The paper is written in a scientifically sound style, has an important impact on the clinical practice and it arouses the interest of the ophthalmologists involved in the treatment of ROP. 

Regarding the subgroup of patients who underwent vitrectomy and lensectomy, it would be useful to indicate how has been aphakia managed ?

Given the interest of the subject, the high significance of its content as well as the high quality of the presentation, the above mentioned paper meets the criteria to be published in IJMS.

Round 2

Reviewer 1 Report

The authors have responded to my concerns and I have no further concerns regarding this publication.